# Decomposed Gaussian Processes for Efficient Regression Models with Low Complexity

**DOI:** 10.3390/e27040393

**Published:** 2025-04-07

**Authors:** Anis Fradi, Tien-Tam Tran, Chafik Samir

**Affiliations:** 1Université Lumière Lyon 2, Université Claude Bernard Lyon 1, ERIC, 69007 Lyon, France; 2Faculty of Applied Sciences, International School, Vietnam National University, Hanoi 10000, Vietnam; 3LIMOS CNRS (UMR 6158), University of Clermont Auvergne, 63000 Clermont-Ferrand, France

**Keywords:** regression, computational complexity, Gaussian process, covariance functions, functional data

## Abstract

In this paper, we address the challenges of inferring and learning from a substantial number of observations (N≫1) with a Gaussian process regression model. First, we propose a flexible construction of well-adapted covariances originally derived from specific differential operators. Second, we prove its convergence and show its low computational cost scaling as $O(Nm^{2})$ for inference and $O(m^{3})$ for learning instead of $O(N^{3})$ for a canonical Gaussian process where N≫m. Moreover, we develop an implementation that requires less memory $O(m^{2})$ instead of $O(N^{2})$. Finally, we demonstrate the effectiveness of the proposed method with simulation studies and experiments on real data. In addition, we conduct a comparative study with the aim of situating it in relation to certain cutting-edge methods.

## 1. Introduction

Gaussian processes are powerful and flexible statistical models that have gained significant popularity in different fields: signal processing, medical imaging, data science, machine learning, econometrics, shape analysis, etc. [1,2,3,4,5]. They provide a nonparametric approach for modeling complex relationships and uncertainty estimation in data [6]. The core idea of Gaussian processes is the assumption that any finite set of data points can be jointly modeled as a multivariate Gaussian distribution [7]. Rather than explicit formulations, Gaussian processes allow for the incorporation of prior knowledge and inference of a nonparametric function *f* that generates the Gaussian process for a set ${\left(t_{i},y_{i}\right)}_{i=1}^{N}$ with $y_{i}=f\left(t_{i}\right)+\tau_{i}$; $t_{i}\in I\subset R^{d}$ and noisy measurements $y_{i}\in R$. If *f* is modeled with a Gaussian process prior, then it can be fully characterized by a mean μ and a covariance function *k*, satisfying(1)$$\begin{array}{ccc} \mu (t) & = & E\left(f(t)\right);\;t\in I \end{array}$$(2)$$\begin{array}{ccc} k(t,s) & = & E\left((f(t)-\mu (t))(f(s)-\mu (s))\right);\;t,s\in I. \end{array}$$
The mean function is usually assumed to be zero (μ(t)=0), whereas the covariance k(t,s) provides the dependence between two inputs, *t* and *s*. Gaussian processes can be applied for various tasks, including regression [8], classification [9], and time series analysis [10]. In regression, Gaussian processes can capture complex and non-linear patterns in data, while, in classification, they enable probabilistic predictions and can handle imbalanced datasets [11]. Additionally, Gaussian processes have been successfully employed in optimization, experimental design, reinforcement learning, and more [12].

One of the key advantages of Gaussian processes is their ability to provide a rich characterization of uncertainty. This makes them particularly suitable for applications where robust uncertainty quantification is crucial, such as in decision-making processes or when dealing with limited or noisy data. Significant efforts have been dedicated to the development of asymptotically efficient or approximate computational methods for modeling with Gaussian processes. However, Gaussian processes may also suffer from some computational challenges. When the number of observations *N* increases, the computational complexity for inference and learning grows significantly and incurs an $O(N^{3})$ computational cost, which is unfeasible for many modern problems [13]. Another limitation of Gaussian processes is the memory scaling $O(N^{2})$ in a direct implementation. To address these issues, various approximations and scalable algorithms, such as sparse Gaussian processes [14,15] and variational inference [16], have been developed to make Gaussian processes applicable to larger datasets.

Certain approximations, as demonstrated in [17,18], involve reduced-rank Gaussian processes that rely on approximating the covariance function. For example, Ref. [19] addressed the computational challenge of working with large-scale datasets by approximating the covariance matrix, which is often required for computations involving kernel methods. In addition, Ref. [20] proposed an FFT-based method for stationary covariances as a technique that leverages the Fast Fourier Transform (FFT) to efficiently compute and manipulate covariance functions in the frequency domain. The link between state space models (SSMs) and Gaussian processes was explored in [21]. This could avoid the cubic complexity in time using Kalman filtering inference methods [22]. Recently, Ref. [23] presented a novel method for approximating covariance functions as an eigenfunction expansion of the Laplace operator defined on a compact domain. More recently, Ref. [24] introduced a reduced-rank algorithm for Gaussian process regression with a numerical scheme.

In this paper, we consider a specific Karhunen–Loève expansion of a Gaussian process with many advantages over other low-rank compression techniques [25]. Initially, we express a Gaussian process as a series of basis functions and random coefficients. By selecting a subset of the most vital basis functions based on the dominant eigenvalues, the rank of the Gaussian process can be reduced. This proves especially advantageous when working with extensive datasets as it can alleviate computational and storage demands. By scrutinizing the eigenvalues linked to the eigenfunctions, one can gauge each eigenfunction’s contribution to noise. This insight can be used for noise modeling, estimation, and separation. The Karhunen–Loève expansion offers a natural framework for model selection and regularization in Gaussian process modeling. By truncating the decomposition to a subset of significant eigenfunctions, we can prevent overfitting. This regularization has the potential to enhance the Gaussian process generalization capability and mitigate the influence of noise or irrelevant features.

In contrast to conventional Karhunen–Loève expansions where eigenfunctions are derived directly from the covariance function or the integral operator, our approach involves differential operators in which the corresponding orthogonal polynomials serve as eigenfunctions. This choice holds significant importance because polynomials are tailored to offer numerical stability and good conditioning, resulting in more precise and stable computations, particularly when dealing with rounding errors. Additionally, orthogonal polynomials frequently possess advantageous properties for integration and differentiation. These properties streamline efficient calculations involving interpolated functions, making them exceptionally valuable for applications necessitating complex computations. On the whole, decomposing Gaussian processes using orthogonal polynomials provides benefits such as numerical stability, quicker convergence, and accurate approximations [26]. However, their incorporation within the Gaussian process framework of the machine learning community has been virtually non-existent. The existing research predominantly revolves around the analysis of integral operators and numerical approximations to compute Karhunen–Loève expansions.

The paper is structured as follows. The first three sections review the necessary theoretical foundations: Section 2 introduces operators in Hilbert spaces, the expansion theorem, and the general Karhunen–Loève theorem; Section 3 focuses on the Gaussian process case; and Section 4 highlights the challenges of canonical Gaussian process regression. Section 5 explores low-complexity Gaussian processes and their computational advantages. Section 6 presents the proposed solutions for various differential operators using orthogonal polynomial bases. Finally, Section 7 discusses the experimental results, followed by a comprehensive discussion and conclusion in Section 8.

## 2. Operators on Hilbert Spaces

In this section, we recall and prove some useful results about linear compact symmetric operators on the particular $L^{2}$ Hilbert space. More general results of any Hilbert space *H* are moved to Appendix A.

Let (Ω,F,P) be a probability space, and let *X* and *Y* be second-order real-valued random variables, meaning $E(X^{2})<\infty$ and $E(Y^{2})<\infty$. By the Cauchy–Schwarz inequality, E(XY)<∞, allowing us to center the variables, assuming without loss of generality that they have zero mean. Thus, $E(X^{2})$ is the variance of *X* and E(XY) is the covariance of *X* and *Y*. The set of these random variables forms a Hilbert space with inner product E(XY) and norm ${\left||X|\right|}_{2}=\sqrt{E(X^{2})}$, leading to mean square convergence: $X_{n}$ converges in norm to *X* if $\left|\right|X_{n}-X{\left|\right|}_{2}\to 0$; equivalently, $E({\left(X_{n}-X\right)}^{2})\to 0$ as n→∞. *X* and *Y* are orthogonal, written X⊥Y, if E(XY)=0. If X⊥Y, then $E{\left(X+Y\right)}^{2}=E\left(X^{2}\right)+E\left(Y^{2}\right)$. For mutually orthogonal variables $X_{1},\ldots ,X_{n}$, we have $E{\left(X_{1}+\cdots +X_{n}\right)}^{2}=E\left(X_{1}^{2}\right)+\cdots +E\left(X_{n}^{2}\right)$.

**Lemma** **1.**
*Let $X={\left\{X_{t}\right\}}_{t}$, $t\in I\subset R^{d}$ be a real-valued second-order random process with zero mean and covariance function $k\left(t,s\right):=E\left(X_{t}X_{s}\right)$. Then, the covariance k is a symmetric non-negative definite function.*


**Proof.** First, by definition of $k\left(t,s\right):=E\left(X_{t}X_{s}\right)$, its symmetry is trivial. Next, it holds that, for all possible choices of $t_{1},\ldots ,t_{N}\in I$, $N\in N^{*}$ and all possible functions ϕ:I→R, we have $$\begin{array}{ccc} \sum_{i=1}^{N}\sum_{l=1}^{N}\phi \left(t_{i}\right)k\left(t_{i},t_{l}\right)\phi \left(t_{l}\right) & = & \sum_{i=1}^{N}\sum_{l=1}^{N}\phi \left(t_{i}\right)E\left(X_{t_{i}}X_{t_{l}}\right)\phi \left(t_{l}\right),\\ & = & E\left(\sum_{i=1}^{N}\sum_{l=1}^{N}\phi \left(t_{i}\right)X_{t_{i}}X_{t_{l}}\phi \left(t_{l}\right)\right),\\ & = & E\left({\left(\sum_{i=1}^{N}\phi \left(t_{i}\right)X_{t_{i}}\right)}^{2}\right)\geq 0. \end{array}$$
Thus, *k* is non-negative definite.    □

In this section, we are interested in the Hilbert space $H:=L^{2}\left(I,\rho \right)$, the space of all the real-valued Borel-measurable functions ϕ on the interval $I\subset R^{d}$ such that $\int_{I}\phi^{2}\left(t\right)\rho \left(t\right)dt<\infty$ with a positive weight function ρ(t), which inherits all proprieties from Appendix A. Consider the Hilbert–Schmidt integral operator $K:L^{2}\left(I,\rho \right)\mapsto L^{2}\left(I,\rho \right)$, expressed as(3)$$\begin{array}{c} \left(K\phi \right)\left(t\right):=\int_{I}k\left(t,s\right)\phi \left(s\right)\rho \left(s\right)ds. \end{array}$$

**Theorem** **1** (Mercer’s Theorem)**.**
*Let k:I×I→R be continuous symmetric non-negative definite and let K be the corresponding Hilbert–Schmidt operator. Let ${\left\{\phi_{j}\right\}}_{j}$ be an orthonormal basis for the space spanned by the eigenvectors corresponding to the non-zero eigenvalues of K. If $\phi_{j}$ is the eigenvector corresponding to the eigenvalue $\lambda_{j}$, then*

(4)
$$k\left(t,s\right)=\sum_{j=1}^{\infty }\lambda_{j}\phi_{j}\left(t\right)\phi_{j}\left(s\right),$$

*where*
*(i)* 
*the series converges absolutely in both variables jointly.*
*(ii)* 
*the series converges to k(t,s) uniformly in both variables jointly.*
*(iii)* 
*the series converges to k(t,s) in $L^{2}\left(I\times I,\rho \right)$.*



**Theorem** **2** (Karhunen–Loève Theorem)**.**
*For a real-valued second-order random process $X={\left\{X_{t}\right\}}_{t}$ with zero mean and continuous covariance function k(t,s) on $I\subset R^{d}$, we can decompose each $X_{t}$ as*

(5)
$$X_{t}=\sum_{j=1}^{\infty }a_{j}\phi_{j}\left(t\right),$$

*where*
*(i)* 
*${\left\{\phi_{j}\right\}}_{j}$ are eigenfunctions of K, which form an orthonormal basis, i.e., $\int_{I}\phi_{j}\left(t\right)\phi_{l}\left(t\right)\rho \left(t\right)dt=\delta_{jl}$, where $\delta_{jl}$ is the Kronecker’s delta.*
*(ii)* 
*$a_{j}=\int_{I}X_{t}\phi_{j}\left(t\right)\rho \left(t\right)dt$ is the coefficient given by the projection of $X_{t}$ onto the j-th deterministic element of the Karhunen–Loève basis in $L^{2}\left(\Omega ,F,P\right)$.*
*(iii)* 
*${\left\{a_{j}\right\}}_{j}$ are pairwise orthogonal random variables with zero mean and variance $\lambda_{j}$, corresponding to the eigenvalue of the eigenfunction $\phi_{j}$.*

*Moreover, the series $\sum_{j=1}^{\infty }a_{j}\phi_{j}\left(t\right)$ converges to $X_{t}$ in mean square uniformly for all t∈I.*


**Proof.** We have $E(a_{j})=0$ and $E\left(a_{j}a_{l}\right)=\lambda_{l}\delta_{jl}$ due to orthonormality of ${\left\{\phi_{j}\right\}}_{j}$. Thus, ${\left\{a_{j}\right\}}_{j}$ are pairwise orthogonal in $L^{2}\left(\Omega ,F,P\right)$ and $v\left(a_{j}\right)=\lambda_{j}$. To show the mean square convergence, let $S_{t}^{m}:=\sum_{j=1}^{m}a_{j}\phi_{j}\left(t\right)$ and $\varepsilon_{t}^{m}:=S_{t}^{m}-X_{t}$. Then,$$\begin{array}{ccc} E({\left(\varepsilon_{t}^{m}\right)}^{2}) & = & \sum_{j=1}^{m}\lambda_{j}\phi_{j}^{2}\left(t\right)-2\sum_{j=1}^{m}\phi_{j}\left(t\right)E\left(a_{j}X_{t}\right)+k\left(t,t\right),\\ & = & k\left(t,t\right)-\sum_{j=1}^{m}\lambda_{j}\phi_{j}^{2}\left(t\right)\underset{m\to \infty }{\to }0, \end{array}$$
uniformly in t∈I by Mercer’s theorem.    □

**Definition** **1.**
*The mean square error is defined as $E({\left(\varepsilon_{t}^{m}\right)}^{2})$. The mean integrated square error (MISE), denoted by $\xi_{m}$, is then given by $\xi^{m}:=\int_{I}E\left({\left(\varepsilon_{t}^{m}\right)}^{2}\right)\rho \left(t\right)dt$, representing the mean square error integrated over the basis ${\left\{\phi_{j}\right\}}_{j}$ onto which $X_{t}$ is projected for every t∈I.*


**Proposition** **1.**
*The MISE $\xi^{m}$ tends to 0 as m→∞.*


**Proof.** The MISE of $S_{t}^{m}$ satisfies$$\begin{array}{ccc} \xi^{m} & = & \int_{I}E\left({\left(\varepsilon_{t}^{m}\right)}^{2}\right)\rho \left(t\right)dt,\\ & = & E(\int_{I}(\sum_{j=m+1}^{\infty }a_{j}\phi_{j}{\left(t)\right)}^{2}\rho \left(t\right)dt),\\ & = & \sum_{j=m+1}^{\infty }E\left(a_{j}^{2}\right),\int_{I}\phi_{j}{\left(t\right)}^{2}\rho \left(t\right)dt,\\ & = & \sum_{j=m+1}^{\infty }E\left(a_{j}^{2}\right),\\ & = & \sum_{j=m+1}^{\infty }\lambda_{j}, \end{array}$$
which tends to 0 as m→∞ since $\lambda_{j}$ are absolutely summable.    □

Now, we highlight the crucial role of the Karhunen–Loève basis in minimizing the error incurred by truncating the expansion of $X_{t}$. By aligning the basis functions with the dominant modes of variation captured by the process, we achieve an optimal representation in terms of mean integrated square error.

**Proposition** **2.**
*The MISE $\xi^{m}$ is minimized if and only if ${\left\{\phi_{j}\right\}}_{j}$ constitutes an orthonormalization of the eigenfunctions of the Fredholm equation*

(6)
(Kϕ)(t)=λϕ(t),

*with ${\left\{\phi_{j}\right\}}_{j}$ arranged to correspond to the eigenvalues ${\left\{\lambda_{j}\right\}}_{j}$ in decreasing magnitude: $\lambda_{1}>\lambda_{2}>\cdots >0$.*


## 3. Expansion and Convergence of Gaussian Processes

A stochastic process $f={\left\{f\left(t\right)\right\}}_{t}$ is said to be a Gaussian process (GP) if for all positive integers $N\in N^{*}$ and all choices of $t_{1},\ldots ,t_{N}$ the random variables $f\left(t_{1}\right),\ldots ,f\left(t_{N}\right)$ form a Gaussian random vector, which means that they are jointly Gaussian. One of the main advantages of a GP is that it can be represented as a series expansion involving a complete set of deterministic basis functions with corresponding random Gaussian coefficients.

**Theorem** **3.**
*If $f={\left\{f\left(t\right)\right\}}_{t}$, $t\in I\subset R^{d}$ is a zero-mean GP of covariance k(t,s) denoted f(t)∼GP(0,k(t,s)), then the Karhunen–Loève expansion projections $a_{j}$ are independent Gaussian random variables: $a_{j}∼N\left(0,\lambda_{j}\right)$.*


Now, we provide a result that establishes important convergence properties, connecting mean square convergence, convergence in probability, and convergence almost surely for sequences of random variables.

**Lemma** **2.**
*1.  The mean square convergence of any sequence ${\left\{a_{n}\right\}}_{n}$ of real-valued random variables implies its convergence in probability.*
*2.* 
*If ${\left\{a_{n}\right\}}_{n}$ is a sequence of independent real-valued random variables, then the convergence of the series $\sum_{n=1}^{\infty }a_{n}$ in probability implies its convergence almost surely.*



**Corollary** **1.**
*For each t∈I, $\sum_{j=1}^{\infty }a_{j}\phi_{j}\left(t\right)$ converges to f(t) almost surely.*


**Proof.** Since ${\left\{a_{j}\right\}}_{j}$ are independent, ${\left\{a_{j}\phi_{j}\left(t\right)\right\}}_{j}$ are also independent because the eigenfunctions are deterministic. Thus, Lemma 2 can be applied to the series of independent random variables $\sum_{j=1}^{\infty }a_{j}\phi_{j}\left(t\right)$. This means that its convergence in probability implies almost sure convergence. Therefore, we only need to prove convergence in probability. This argument is straightforward because, by Lemma 2, convergence in mean square implies convergence in probability. The mean square convergence of the partial sums $f^{m}\left(t\right):=\sum_{j=1}^{m}a_{j}\phi_{j}\left(t\right)$ is a result of the Karhunen–Loève theorem.    □

**Example** **1.**
*1.  The Karhunen–Loève expansion of the Brownian motion on I=[0,1] as a centered GP with covariance k(t,s)=min(t,s) is given by*

$$f\left(t\right)=\frac{2\sqrt{2}}{\pi }\sum_{j=1}^{\infty }\frac{a_{j}^{*}}{2j-1}\sin\left(\frac{(2j-1)\pi }{2}t\right),$$

*where $a_{j}^{*}=\frac{a_{j}}{\sqrt{\lambda_{j}}}$ and $\lambda_{j}=\frac{4}{{\left(2j-1\right)}^{2}\pi^{2}}$.*
*2.* 
*The Karhunen–Loève expansion of the Brownian bridge on I=[0,1] as a centered GP with covariance k(t,s)=min(t,s)−ts is given by*

$$f\left(t\right)=\frac{\sqrt{2}}{\pi }\sum_{j=1}^{\infty }\frac{a_{j}^{*}}{j}\sin\left(j\pi t\right),$$


*where $a_{j}^{*}=j\pi a_{j}$ and $\lambda_{j}=\frac{1}{j^{2}\pi^{2}}$.*



## 4. Ill-Conditioned Canonical Gaussian Process Regression

In a regression task, a nonparametric function *f* is assumed to be a realization of a stochastic GP prior, whereas the likelihood term holds from observations corrupted by a noise term according to the canonical form(7)$$\begin{array}{cc} y_{i} & =f\left(t_{i}\right)+\tau_{i};\;i=1,\ldots ,N,\\ f(t) & ∼\mathrm{GP}(0,k(t,s)), \end{array}$$
where $\tau_{i}∼N\left(0,\sigma_{N}^{2}\right)$ is a Gaussian noise. Given a training dataset $D=\left(t,y\right)={\left(t_{i},y_{i}\right)}_{i=1}^{N}$, the posterior distribution over $f=f\left(t\right)={\left(f\left(t_{1}\right),\ldots ,f\left(t_{N}\right)\right)}^{⊤}$ is also Gaussian: P(f|D)=N(μ,Σ). From Bayes’ rule, we state that the mean and the covariance posterior are expressed as(8)$$\begin{array}{ccc} \mu & = & K{\left(K+\sigma_{N}^{2}I_{N}\right)}^{-1}y, \end{array}$$(9)$$\begin{array}{ccc} \Sigma & = & {\left(K^{-1}+\frac{1}{\sigma_{N}^{2}}I_{N}\right)}^{-1}, \end{array}$$
where $K={\left[k(t_{i},t_{j})\right]}_{i,j=1}^{N}$ is the prior covariance matrix and $I_{N}$ is the N×N identity matrix. The predictive distribution at any test input $t_{★}$ can be computed in closed form as $f\left(t_{★}\right)|D,t_{★}∼N\left({\bar{f}}_{★},v\left(f_{★}\right)\right),$ with(10)$$\begin{array}{cc} {\bar{f}}_{★} & =k{\left(t_{★}\right)}^{⊤}{\left(K+\sigma_{N}^{2}I_{N}\right)}^{-1}y, \end{array}$$(11)$$\begin{array}{cc} v(f_{★}) & =k\left(t_{★},t_{★}\right)-k{\left(t_{★}\right)}^{⊤}{\left(K+\sigma_{N}^{2}I_{N}\right)}^{-1}k\left(t_{★}\right), \end{array}$$
where $k\left(t_{★}\right)={\left[k(t_{i},t_{★})\right]}_{i=1}^{N}$.

The covariance function *k* usually depends on a set of hyperparameters denoted $\theta_{k}$ that need to be estimated from the training dataset. The log marginal likelihood for GP regression serves as an indicator of the degree to which the selected model accurately captures the observed patterns. The log marginal likelihood is typically used for model selection and optimization. Let $\theta =(\theta_{k},\sigma_{N}^{2})$ denote the set of all model parameters, and then the log marginal likelihood logP(y|t,θ) is given by(12)$$\begin{array}{c} l\left(\theta \right)=-\frac{1}{2}\log\left|K+\sigma_{N}^{2}I_{N}\right|-\frac{1}{2}y^{⊤}{\left(K+\sigma_{N}^{2}I_{N}\right)}^{-1}y-\frac{N}{2}\log\left(2\pi \right). \end{array}$$
Here, |.| denotes the determinant. The goal is to estimate θ that maximizes the log marginal likelihood. This can be achieved using different methods, such as gradient-based algorithm [27]. The weakness of inferring the posterior mean or the mean prediction or even learning the hyperparameters from the log marginal likelihood is the need to invert the N×N Gram matrix $K+\sigma_{N}^{2}I_{N}$. This operation costs $O(N^{3})$, which limits the applicability of standard GPs when the sample size *N* increases significantly. Furthermore, the memory requirements for GP regression scale with a computational complexity of $O(N^{2})$.

A covariance function k(t,s) is said to be stationary (isotropic) if it is invariant to translation, i.e., a function of ||t−s|| only. Two commonly used stationary covariance functions for GP regression are the Squared Exponential (SE) and Matérn-ν kernels defined by(13)$$\begin{array}{ccc} k(t,s) & = & \sigma^{2}e^{-\varepsilon^{2}{∥t-s∥}^{2}};\;t,s\in R \end{array}$$(14)$$\begin{array}{ccc} k(t,s) & = & \sigma^{2}\frac{2^{1-\nu }}{\Gamma (\nu )}{\left(\varepsilon \sqrt{2\nu }∥t-s∥\right)}^{\nu }K_{\nu }\left(\varepsilon \sqrt{2\nu }∥t-s∥\right);\;t,s\in R \end{array}$$
respectively, where $\sigma^{2}$ is the variance parameter controlling the amplitude of the covariance, ε is the shape parameter, and ν=k+1/2; k∈N is the half integer smoothness parameter controlling its differentiability. Here, Γ is the gamma function and $K_{\nu }$ is the modified Bessel function of the second kind. Both the SE and Matérn covariance functions have hyperparameters that needs to be estimated from the data during the model training process. A GP with a Matérn-ν covariance is ⌈ν⌉−1 times differentiable in the mean-square sense. The SE covariance is the limit of Matérn-ν as the smoothness parameter ν approaches infinity. When choosing between the SE and Matérn covariance functions, it is often a matter of balancing the trade-off between modeling flexibility and computational complexity. The SE covariance function is simpler and more computationally efficient but may not capture complex patterns in data as well as the Matérn covariance function with an appropriate choice of smoothness parameter.

In order to compute (Equation 10), we also need to invert the N×N Gram matrix $K+\sigma_{N}^{2}I_{N}$. This task is impractical when the size sample *N* is large because inverting the matrix leads to $O(N^{2})$ memory and $O(N^{3})$ time complexities [28]. There are several methods to overcome this difficulty. For instance, the variational inference proceeds by introducing *n* inducing points and corresponding *n* inducing variables. The variational parameters of inducing variables are learned by minimizing the Kullback–Leibler divergence [16]. Picking n≪N, the complexity reduces to $O(Nn^{2}+n^{3})$ in prediction and $O(n^{2})$ in minimizing the Kullback–Leibler divergence. The computational complexity of conventional sparse Gaussian process (SGP) approximations typically scales as $O(Nn^{2})$ in time for each step of evaluating the marginal likelihood [14]. The storage demand scales as O(Nn). This arises from the unavoidable cost of re-evaluating all results involving the basis functions at each step and the need to store the matrices required for these calculations.

## 5. Low-Complexity Gaussian Process Regression

In order to avoid the inversion of the N×N Gram matrix $K+\sigma_{N}^{2}I_{N}$, we use the approximation scheme presented in Section 3 and rewrite the GP with a truncated set of *m* basis functions. Hence, the truncated *f* at an arbitrary order $m\in N^{*}$ is given by(15)$$\begin{array}{c} f^{m}\left(t\right):=\sum_{j=1}^{m}a_{j}\phi_{j}\left(t\right), \end{array}$$
with an approximation error $e^{m}\left(t\right):=\sum_{j=m+1}^{\infty }a_{j}\phi_{j}\left(t\right)$. The canonical GP regression model (Equation 7) becomes(16)$$\begin{array}{cc} y_{i} & =f^{m}\left(t_{i}\right)+\tau_{i};\;i=1,\ldots ,N,\\ f^{m}\left(t\right) & ∼\mathrm{GP}(0,k^{m}\left(t,s\right)), \end{array}$$
with a covariance function approximated by $k^{m}\left(t,s\right)=\sum_{j=1}^{m}\lambda_{j}\phi_{j}\left(t\right)\phi_{j}\left(s\right)$. The convergence degree of the Mercer series in (Equation 4), that is, $k^{m}\underset{m\to \infty }{\to }k$ with $k\left(t,s\right)=\sum_{j=1}^{\infty }\lambda_{j}\phi_{j}\left(t\right)\phi_{j}\left(s\right),$ depends heavily on the eigenvalues and the differentiability of the covariance function. Ref. [29] showed that the speed of the uniform convergence varies in terms of the decay rate of eigenvalues and demonstrated that, for a 2β times differentiable covariance *k*, the truncated covariance $k^{m}$ approximates *k* as $O({\left(\sum_{j=m+1}^{\infty }\lambda_{j}\right)}^{\frac{\beta }{\beta +1}})$. For infinitely differentiable covariances, the latter is $O({\left(\sum_{j=m+1}^{\infty }\lambda_{j}\right)}^{1-\epsilon })$ for any ϵ>0. To summarize, smoother covariance functions tend to exhibit faster convergence, while less smooth or non-differentiable covariance functions may exhibit slower or no convergence.

The resulting covariance falls into the class of reduced-rank approximations based on approximating the covariance matrix K with a matrix $\tilde{K}={\left[k^{m}\left(t_{i},t_{j}\right)\right]}_{i,j=1}^{N}=\Phi \Gamma \Phi^{⊤}$, where Γ is an m×m diagonal matrix with eigenvalues such that $\Gamma_{jj}=\lambda_{j}$ and Φ is an N×m matrix with eigenfunctions such that $\Phi_{ij}=\phi_{j}\left(t_{i}\right)$. Note that the approximate covariance matrix $\tilde{K}$ becomes ill-conditioned when the ratio $\lambda_{1}/\lambda_{m}$ is large. This ill-conditioning occurs particularly when the observation points $t_{i}$ are too close to each other [30]. In practice, this can lead to significant numerical errors when inverting $\tilde{K}$, resulting in unstable solutions, amplified errors in parameter estimation, and unreliable model predictions.

Now, we show how the approximated regression models make use of GP decomposition to achieve low complexity. We write down the expressions needed for inference and discuss the computational requirements. Applying the matrix inversion lemma [31], we rewrite the predictive distribution (Equation 10) and (Equation 11) as(17)$$\begin{array}{cc} {\bar{f}}_{★} & \approx \phi_{★}^{⊤}{\left(\Phi^{⊤}\Phi +\sigma_{N}^{2}\Gamma^{-1}\right)}^{-1}\Phi^{⊤}y, \end{array}$$(18)$$\begin{array}{cc} v(f_{★}) & \approx \sigma_{N}^{2}\phi_{★}^{⊤}{\left(\Phi^{⊤}\Phi +\sigma_{N}^{2}\Gamma^{-1}\right)}^{-1}\phi_{★}, \end{array}$$
where $\phi_{★}$ is an *m*-dimensional vector with the *j*-th entry being $\phi_{j}\left(t_{★}\right)$. When the number of observations is much higher than the number of required basis functions (N≫m), the use of this approximation is advantageous. Thus, any prediction mean evaluation is dominated by the cost of constructing $\Phi^{⊤}\Phi$, which means that the method has an overall asymptotic computational complexity of $O(Nm^{2})$.

The approximate log marginal likelihood satisfies(19)$$\begin{array}{cc} l(\theta )\approx & -\frac{1}{2}\log\left|\Phi \Gamma \Phi^{⊤}+\sigma_{N}^{2}I_{N}\right|-\frac{1}{2}y^{⊤}{\left(\Phi \Gamma \Phi^{⊤}+\sigma_{N}^{2}I_{N}\right)}^{-1}y-\frac{N}{2}\log\left(2\pi \right),\\ = & -\frac{1}{2}\left(N-m\right)\log\sigma_{N}^{2}-\frac{1}{2}\log\left|\Phi^{⊤}\Phi +\sigma_{N}^{2}\Gamma^{-1}\right|-\frac{1}{2}\sum_{j=1}^{m}\log\lambda_{j}\\ \; & -\frac{1}{2\sigma_{N}^{2}}\left(y^{⊤}y-y^{⊤}\Phi {\left(\Phi^{⊤}\Phi +\sigma_{N}^{2}\Gamma^{-1}\right)}^{-1}\Phi^{⊤}y\right)-\frac{N}{2}\log\left(2\pi \right). \end{array}$$
Consequently, evaluating the approximate log marginal likelihood has a complexity of $O(m^{3})$ needed to inverse the m×m matrix $M=\Phi^{⊤}\Phi +\sigma_{N}^{2}\Gamma^{-1}$. In practice, if the sample size *N* is large, it is preferable to store the result of $\Phi^{⊤}\Phi$ in memory, leading to a memory requirement that scales as $O(m^{2})$. For efficient implementation, matrix-to-matrix multiplications can often be avoided, and the inversion of M can be performed using Cholesky factorization for numerical stability. This factorization ($LL^{⊤}=M$) can also be used to compute the term $\log|M|=2\sum_{j=1}^{m}\log L_{jj}$. Algorithm 1 outlines the main steps for estimating the hyperparameters of the low-complexity GP.
**Algorithm 1** Gradient Ascent for Hyperparameter Learning.**Require:** 
Data D=(t,y), initial hyperparameters $\theta =(\theta_{k},\sigma_{N}^{2})$, learning rate η, tolerance ϵ, maximum iterations *T*.**Ensure:** 
Optimized hyperparameters $\theta^{*}$.1:Initialize $\theta ={\left(\theta_{j}\right)}_{j}$.2:**for** t=1 to *T* **do**3:    Construct Φ and Γ depending on θ.4:    Compute $M=\Phi^{⊤}\Phi +\sigma_{N}^{2}\Gamma^{-1}$.5:    Evaluate L=Cholesky(M).6:    **for** each hyperparameter $\theta_{j}$ **do**7:        Compute gradient $g_{j}=\mathrm{function}\left(M\right)$.8:        Update hyperparameter: $\theta_{j}\leftarrow \theta_{j}+\eta g_{j}$.9:    **end for**10:    **if** ∥∇l(θ)∥<ϵ **then**11:        **break**12:    **end if**13:**end for**14:**return** optimized hyperparameters $\theta^{*}=\left(\theta_{k}^{*},\sigma_{N}^{2,*}\right)$.

## 6. Closed Solutions from Differential Operators

In this section, we highlight the connection between Hilbert–Schmidt integral operators in (Equation 3) and differential operators. Thus, we describe explicit solutions for the low complexity GP (LCGP) with covariances derived from differential operators. Unlike previous works where the eigen-decomposition is determined from the Mercer series or the integral operator directly, this paper focuses on constructing covariance functions that incorporate orthogonal polynomials as eigenfunctions. It is worth noting that polynomials can approximate a wide range of functions with various degrees of complexity. They can be adjusted to predict different data patterns and capture both linear and non-linear relationships [32].

The connection between a differential operator denoted L and the integral operator K has been largely used. We follow the same idea in [33,34,35] and define Green’s function *G* of the differential operator L as its “right inverse”, i.e.,(20)$$\begin{array}{c} (LG)(t,s)=\delta (t-s);\;t,s\in I, \end{array}$$
where δ(.) denotes the Dirac delta function.

**Theorem** **4.**
*Let k:I×I→R be continuous symmetric non-negative definite and let K be the corresponding Hilbert–Schmidt operator. Let ${\left\{\phi_{j}\right\}}_{j}$ be an orthonormal basis for the space spanned by the eigenfunctions corresponding to the non-zero eigenvalues of K. If $\phi_{j}$ is the eigenfunction associated with the eigenvalue $\lambda_{j}$ and the covariance function acts as a Green’s function of a differential operator L, then the eigenvalues of K correspond to reciprocal eigenvalues of $\rho {\left(t\right)}^{-1}L$, while the corresponding eigenfunctions are still the same.*


**Proof.** We have$$\begin{array}{ccc} \lambda_{j}\left(L\phi_{j}\right)\left(t\right) & = & L\left(K\phi_{j}\right)\left(t\right),\\ & = & L\int_{I}k\left(t,s\right)\phi_{j}\left(s\right)\rho \left(s\right)ds,\\ & = & \int_{I}Lk\left(t,s\right)\phi_{j}\left(s\right)\rho \left(s\right)ds,\\ & = & \int_{I}\delta \left(t-s\right)\phi_{j}\left(s\right)\rho \left(s\right)ds,\\ & = & \phi_{j}\left(t\right)\rho \left(t\right). \end{array}$$
Finally, we obtain$$\begin{array}{c} \rho^{-1}\left(t\right)\left(L\phi_{j}\right)\left(t\right)=\frac{1}{\lambda_{j}}\phi_{j}\left(t\right), \end{array}$$
which completes the proof. □

At this stage, we compute eigenvalues and eigenfunctions of $\rho {\left(t\right)}^{-1}L$, from which we deduce the Mercer decomposition of k(t,s) given in (Equation 4) replacing $\lambda_{j}$ by $\gamma_{j}=\frac{1}{\lambda_{j}}$. We select a list of some interesting and useful differential operators that act on $L^{2}\left(I,\rho \right)$, for example: Matérn, Legendre, Laguerre, Hermite, Chebyshev, and Jacobi, from which we explicitly find the corresponding eigen-decompositions. Table 1 summarizes each class of L, the index set *I*, the weight function ρ, the eigenvalues $\gamma_{j}$, the eigenfunctions as polynomials $\phi_{j}$, and the resulting MISE. Note that, for Laguerre, Hermite, and Chebyshev polynomials, the eigenvalues $\gamma_{j}$ of L were initially negatives. Therefore, we consider the iterated operator $L^{2}:=L\circ L$ with squared eigenvalues and unchanged eigenfunctions. Further, for Legendre, Hermite, and Chebyshev, we state that $∥\phi_{j}{∥}_{L^{2}}\neq 1$, which means that $\phi_{j}$ should be normalized to form an orthonormal basis. For Jacobi, the differential operator is $L_{J}=\left(t^{2}-1\right)\frac{d^{2}}{dt^{2}}+\left[\alpha -\beta +(\alpha +\beta +2)t\right]\frac{d}{dt}$ with parameters α,β greater than −1. More details about the corresponding eigenfunctions $J_{j}^{\alpha ,\beta }$ and the $L^{2}$-norm are provided in [36].

Figure 1 illustrates the behavior of the eigenvalues $\lambda_{j}=\frac{1}{\gamma_{j}}$ of the integral operator K as the index *j* varies from 1 to 40. This is attributed to the smoothness of the true covariance as *m* is growing. Figure 2 visually depicts several GP realizations across various differential operator settings.

## 7. Experiments

In this section, we assess the effectiveness of the proposed LCGP by conducting evaluations on multiple datasets and reporting comparisons with some state-of-the-art methods. The comparative analysis will enable us to gain insights into the strengths and weaknesses of the proposed framework and determine its competitiveness. Here is an overview of the provided Python (version 3.12.4, packaged by Anaconda) code (https://github.com/anisfradi/Low-Complexity-Regression-Models-with-Gaussian-Process-Prior.git, accessed on 3 April 2025). The computations were executed on a computer with 125 GB of memory and a Xeon(R) W 2275 CPU @ 3.30 GHz. Throughout all experiments, we set the truncation parameter *m* to 25.

### 7.1. Simulations

In simulation study, we examine two functions: a parametric beta density function represented by f(t)=B(t|a=2,b=5) (Simulation 1) and a nonparametric quasi-periodic function satisfying f(t)=tsin(10t) (Simulation 2). Both functions are defined on the unit interval I=[0,1]. For these experiments, we generated a total of 140 observations. Out of these, we allocated 40 observations for training and the remaining for testing. The input points $t_{i}$ are uniformly distributed on [0,1]. To introduce variability and simulate real-world conditions, each observed point was calculated as $y_{i}=f\left(t_{i}\right)+\tau_{i}$, where $\tau_{i}$ represents Gaussian noise drawn from $N(0,\sigma_{N}^{2}=0.1)$. This procedure allows us to evaluate the performance of our models using noisy data.

In Figure 3, we show an illustration of predicting the true function from simulations. We observe that different types of polynomial eigenfunctions have distinct advantages in prediction with truncated GP. Matérn (MGP) and Legendre (LGP) are well suited for functions with exponential decay, making them suitable for decay processes. They also incorporate function values at data points, allowing for accurate predictions even with rapidly changing functions and effective in approximating functions with oscillatory behavior. Hermite (HGP) and Chebyshev (CGP) are accurate with more slowly varying processes. Jacobi (JGP) accurately fits functions passing through given data points. The performance of the Laguerre operator is comparatively weaker, leading to its exclusion from the experimental analysis. Figure 4 illustrates the uncertainty level of the proposed MGP when applied to the predicted data.

### 7.2. Real Data

In this part, we conduct a real study using two challenging datasets. The first dataset comprises more than 10,000 observations collected by the California Cooperative Oceanic Fisheries Investigations (CalCOFI) (https://www.kaggle.com/datasets/sohier/calcofi?resource=download, accessed on 3 April 2025). It investigates the ecological aspects surrounding the collapse of the sardine population of the coast of California, which is recognized as the longest and most comprehensive time series of oceanographic and larval fish data worldwide. It encompasses abundance data for over 250 fish species’ larvae, as well as larval length frequency data, egg abundance data for important commercial species, and oceanographic data. Data collected at depths up to 500 meters include temperature, salinity, oxygen, phosphate, silicate, nitrate and nitrite, chlorophyll, phytoplankton biodiversity, etc. In this experiment, we are specifically targeting climate change indicators on the California coast when we focus on data illustrating the temperature (°C) as function of the salinity (ppt). Some examples (N=500) are given in Figure 5 (left).

The second dataset used in this study pertains to Medical Cost Personal (mCP) (https://www.kaggle.com/datasets/mirichoi0218/insurance/discussion, accessed on 3 April 2025, which was sourced from demographic statistics provided by the US Census Bureau. It primarily focuses on the cost of treatment, which is influenced by various factors, including age, sex, body mass index (BMI), and smoking status. Specifically, this paper examines the relationship between treatment costs (charges in thousand dollars) and the BMI factor for both smokers and non-smokers. See some examples for smokers (N=532) and non-smokers (N=137) in Figure 5 (right). For both real datasets, a random split of 50% is designated for training, with the remaining for testing. This strategy ensures a balanced distribution of data for model training and facilitates a comprehensive assessment of model performance. We compare our proposed LCGP with several baseline methods to determine if there are significant performance differences. The baseline methods include (i) the standard GP (std-GP) with a Matérn covariance function (as described in Section 4 and (ii) the sparse GP (SGP) [14]. The standard GP model was implemented in the scikit-learn library (https://scikit-learn.org/stable/modules/generated/sklearn.gaussian_process.GaussianProcessRegressor.html, accessed on 3 April 2025), and the sparse GP was implemented in GPy (https://nbviewer.org/github/SheffieldML/notebook/blob/master/GPy/sparse_gp_regression.ipynb, accessed on 3 April 2025). To evaluate the performance of the proposed methods, two commonly used metrics include the following:ISE: The integrated squared error as the average squared difference between the predicted function and the true function.R-squared: The coefficient of determination as a statistical measure used in regression analysis that represents the proportion of the variance in the dependent variable that is predictable from the independent variable.

Table 2, Table 3 and Table 4 present the prediction criteria obtained from real data and the computational time needed for this evaluation in seconds (s). Notably, Legendre exhibits slightly superior performance compared to other methods on CalCOFI and mCP smoker datasets, while Matérn emerges as the top performer on mCP non-smoker data. The number of inducing points is fixed to n=10<m for the sparse GP. These results suggest that Legendre demonstrates greater adaptability in capturing a diverse array of patterns and structures within real data. Despite its equivalent complexity to other proposed models, the Chebyshev polynomial exhibits the shortest computational time. This efficiency can be attributed to a faster evaluation of the corresponding eigenfunctions. For further insights into the quality of predictions, refer to Figure 6 and Figure 7.

We also estimate the number of operations required for each approach. The sparse GP typically involves a computational complexity of $O(N\times 10^{2})$, where *N* is the number of training points and n=10 the number of inducing points. This results in approximately $c_{1}\times N\times 10^{2}$ operations, where $c_{1}$ is a constant depending on matrix operations. In contrast, our LCGP method operates with a complexity of $O(N\times 25^{2})$, leading to $c_{2}\times N\times 25^{2}$ operations, where m=25 is the number of basis functions and $c_{2}$ a corresponding constant.

To assess the robustness of our method, we have conducted a series of experiments on CalCOFI dataset by gradually increasing the sample size *N* to 12,000. The results shown in Figure 8 demonstrate that our methods are highly competitive in terms of accuracy: low ISE and high R-squared values. However, it is worth noting that the gap in computational time (measured in minutes) between the methods becomes significant.

## 8. Conclusions

In this paper, we have introduced a novel regression model with a Gaussian process prior. This nonparametric model is designed for inferring, predicting, and learning. Contrary to previous methods, the proposed one is derived from specific differential operators. We study and test different configurations with well adapted eigenfunctions’ bases, enabling straightforward implementations with closed-form expressions. In summary, a key advantage of our method is its ability to overcome the limitations associated with standard Gaussian processes. This includes reducing the computational cost to $O(Nm^{2})$ for inference and $O(m^{3})$ for learning. We assess the effectiveness of our proposed model using a variety of simulated and real-world data. The experimental results and comparisons demonstrate its high accuracy, low computational overhead, and analytical simplicity in comparison to the existing methods.

## Figures and Tables

**Figure 1 entropy-27-00393-f001:**
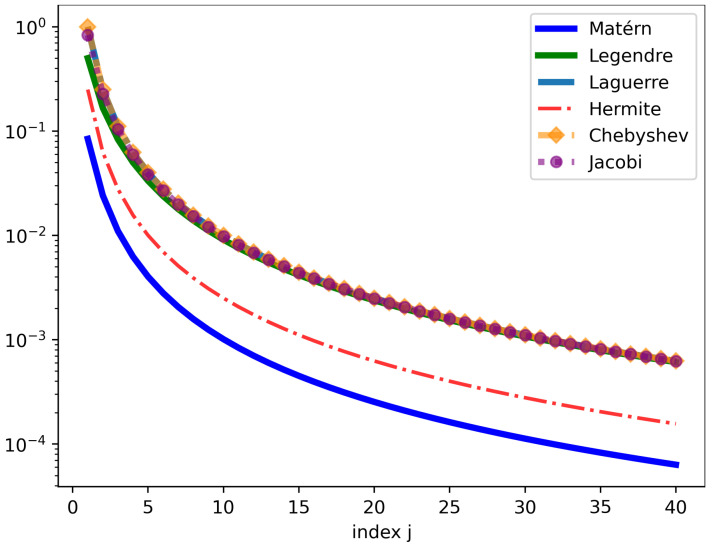
The eigenvalues $\lambda_{j}=\frac{1}{\gamma_{j}}$ for different differential operators using a base-logarithmic scale with $\sigma^{2}=2,\varepsilon =2$, α=1 for Matérn and α=−0.5,β=−0.3 for Jacobi.

**Figure 2 entropy-27-00393-f002:**
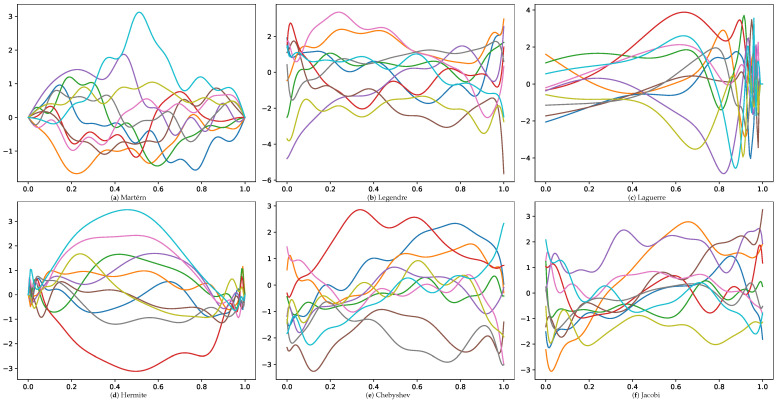
Ten sample realizations from GPs corresponding to different differential operators. Each subfigure represents a distinct operator, with 10 colored curves illustrating different GP realizations.

**Figure 3 entropy-27-00393-f003:**
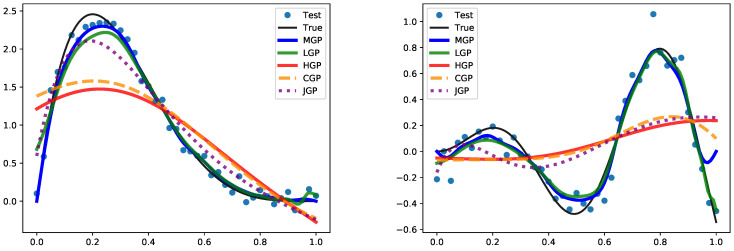
The prediction with LCGP regarding Simulation 1 (**left**) and Simulation 2 (**right**).

**Figure 4 entropy-27-00393-f004:**
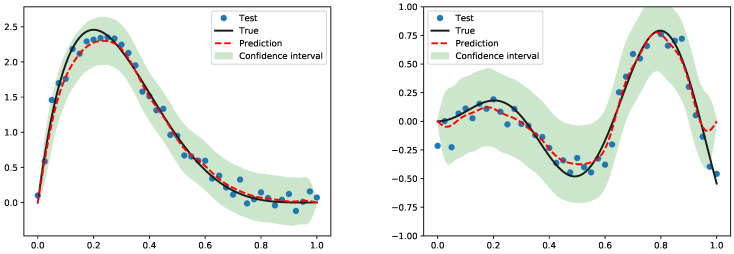
The uncertainty with MGP regarding Simulation 1 (**left**) and Simulation 2 (**right**).

**Figure 5 entropy-27-00393-f005:**
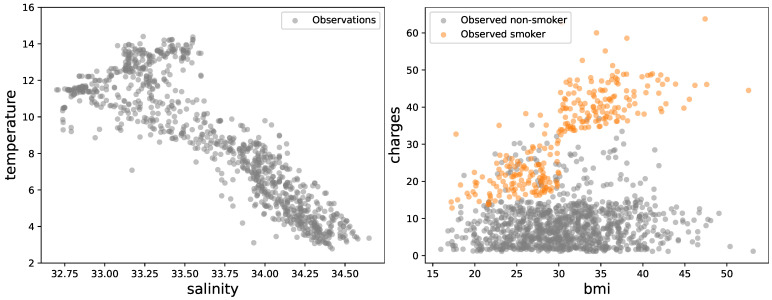
Some observations from real datasets: CalCOFI (**left**) and mCP (**right**).

**Figure 6 entropy-27-00393-f006:**
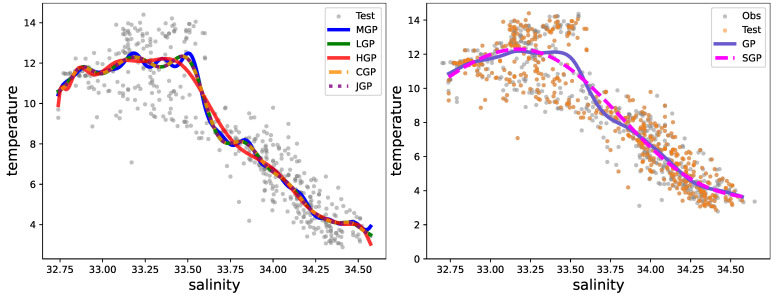
The prediction regarding CalCOFI with LCGP (**left**) and other methods (**right**).

**Figure 7 entropy-27-00393-f007:**
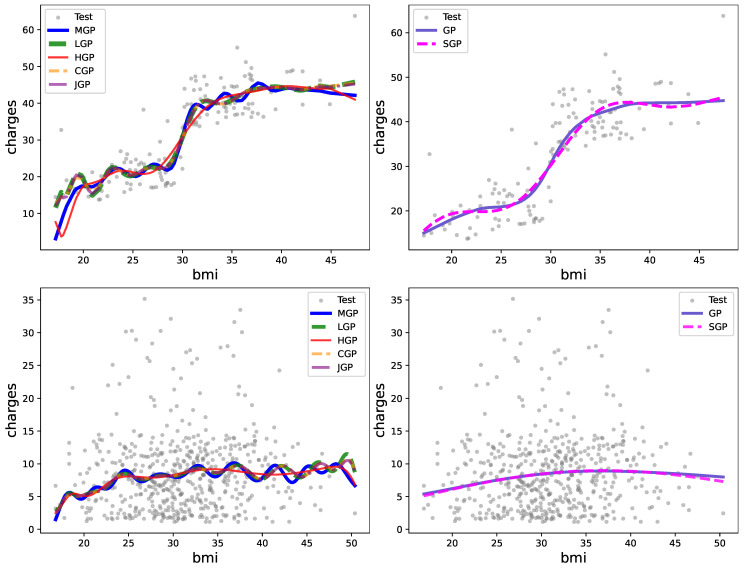
Illustration of the prediction results regarding mCP with the proposed LCGP (**left**) and comparative methods (**right**): (**top**) smoker class and (**bottom**) non-smoker class.

**Figure 8 entropy-27-00393-f008:**
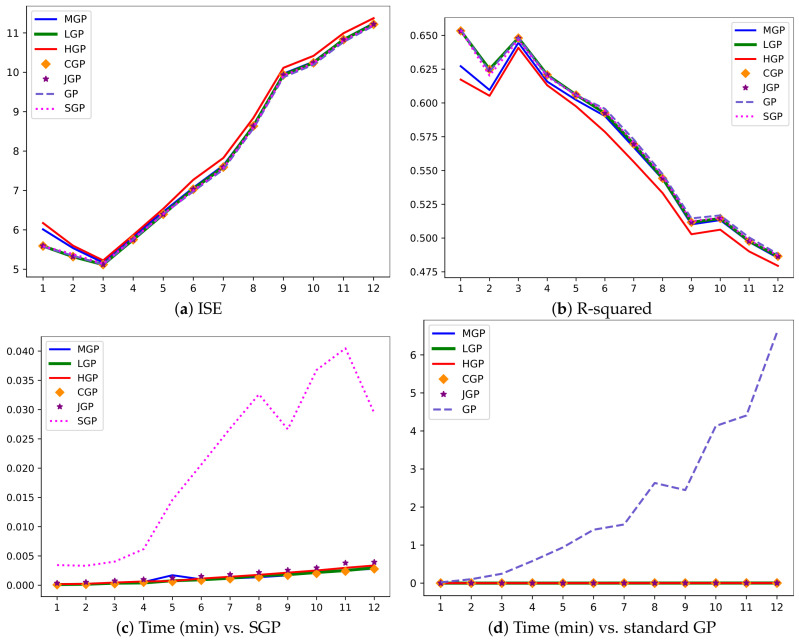
Results regarding CalCOFI dataset with an increasing sample size: N=k×1000, where k=1,…,12.

**Table 1 entropy-27-00393-t001:** Various differential operators and their corresponding decompositions.

Operator	L	*I*	ρ	$\gamma_{j}$	$\phi_{j}\left(t\right)$	$∥\phi_{j}{∥}_{L^{2}}$	MISE
Matérn ^1^	$\sigma^{-2}{\left(\varepsilon -\frac{d^{2}}{dt^{2}}\right)}^{\alpha }$	[0,1]	1	$\sigma^{-2}{\left(\varepsilon +j^{2}\pi^{2}\right)}^{\alpha }$	$\sqrt{2}\sin\left(j\pi t\right)$	1	$\sigma^{2}\sum_{j=m+1}^{\infty }{\left(\varepsilon +j^{2}\pi^{2}\right)}^{-\alpha }$
Legendre	$-\left(1-t^{2}\right)\frac{d^{2}}{dt^{2}}+2t\frac{d}{dt}$	[−1,1]	1	j(j+1)	$\frac{1}{2^{j}j!}\frac{d^{j}}{dt^{j}}{\left(t^{2}-1\right)}^{j}$	$\sqrt{\frac{2}{2j+1}}$	$\frac{1}{m+1}$
Laguerre	${\left(t\frac{d^{2}}{dt^{2}}+\left(1-t\right)\frac{d}{dt}\right)}^{2}$	[0,∞)	$e^{-t}$	$j^{2}$	$\frac{e^{t}}{j!}\frac{d^{j}}{dt^{j}}\left(e^{-t}t^{j}\right)$	1	$\frac{\pi^{2}}{6}-\sum_{j=1}^{m}\frac{1}{j^{2}}$
Hermite	${\left(\frac{d^{2}}{dt^{2}}-2t\frac{d}{dt}\right)}^{2}$	R	$e^{-t^{2}}$	$4j^{2}$	${\left(-1\right)}^{j}e^{t^{2}}\frac{d^{j}}{dt^{j}}e^{-t^{2}}$	$\sqrt{\sqrt{\pi }2^{j}j!}$	$\frac{1}{4}\left(\frac{\pi^{2}}{6}-\sum_{j=1}^{m}\frac{1}{j^{2}}\right)$
Chebyshev	${\left(-\left(1-t^{2}\right)\frac{d^{2}}{dt^{2}}+t\frac{d}{dt}\right)}^{2}$	[−1,1]	$\frac{1}{\sqrt{1-t^{2}}}$	$j^{2}$	cos(jarccost)	$\sqrt{\frac{\pi }{2}}$	$\frac{\pi^{4}}{90}-\sum_{j=1}^{m}\frac{1}{j^{4}}$
Jacobi	$L_{J}$	[−1,1]	${\left(1-t\right)}^{\alpha }{\left(1+t\right)}^{\beta }$	j(j+α+β+1)	$J_{j}^{\alpha ,\beta }\left(t\right)$	see [36]	$\sum_{j=m+1}^{\infty }\frac{1}{j(j+\alpha +\beta +1)}$

^1^ Matérn hyperparameters: $\sigma^{2}$ for variance, ε for shape, and α for smoothness: α=ν+1/2=k+1; k∈N.

**Table 2 entropy-27-00393-t002:** Results on the CalCOFI dataset. Optimal values across all methods are highlighted in red.

Method	ISE	R-Squared	Time (s)
MGP	1.5186	0.8588	0.0048
LGP	1.4988	0.8607	0.0033
HGP	1.5095	0.8597	0.0059
CGP	1.4991	0.8606	0.0022
JGP	1.4992	0.8606	0.0171
std-GP (standard)	1.4997	0.8606	0.2259
SGP (sparse)	1.5635	0.8547	0.1489

**Table 3 entropy-27-00393-t003:** Results on the mCP non-smoke dataset. Optimal values across all methods are highlighted in red.

Method	ISE	R-Squared	Time (s)
MGP	0.2556	−0.0029	0.0042
LGP	0.2568	−0.0076	0.0046
HGP	0.2578	−0.0117	0.0045
CGP	0.2572	−0.0091	0.0023
JGP	0.2570	−0.0086	0.0175
std-GP (standard)	0.2570	−0.0086	0.5880
SGP (sparse)	0.2569	−0.0083	0.1803

**Table 4 entropy-27-00393-t004:** Results on the mCP smoke dataset. Optimal values across all methods are highlighted in red.

Method	ISE	R-Squared	Time (s)
MGP	0.2152	0.7611	0.0018
LGP	0.1809	0.7992	0.0018
HGP	0.2518	0.7205	0.0020
CGP	0.1822	0.7977	0.0011
JGP	0.1836	0.7962	0.0139
std-GP (standard)	0.2018	0.7760	0.0288
SGP (sparse)	0.2227	0.7528	0.1696

## Data Availability

No new data were created or analyzed in this study. Data sharing is not applicable to this article.

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
