# Peer review of "Decomposed Gaussian Processes for Efficient Regression Models with Low Complexity"

_entropy, 2025, doi:10.3390/e27040393_

Round 1

Reviewer 1 Report

Comments and Suggestions for Authors

The authors proposes a new method to approximate the covariance matrix in Gaussian processes for regression (GPR). The resort to the Mercer’s theorem to write the convariance as a combination of basis functions, within the family of low-rank approximations. The approach is based on a truncated, not infinite, expansión of the Mercer’s series. The equations are rewritten so the number of terms, m, is the main dimension of the matrices to invert and hence the computational complexity is reduced. Code is provided.

I find the manuscript quite interesting, both at a theoretical and practical level. There are some low-rank approximations such as the Nystrom or the one with inducing points. But this is, to the best of my knowledge, a novel proposal. Furthermore, as reported, the reduction in computation time is remarkable.

However, I find several issues that should be reviewed before definitely accepting the publication. See below my comments and suggestions:

  • As far as I understand it is not until Section 5 that you introduce any new content. But you introduce even proofs. This is rare for a publication, where usually the second section includes all needed, not new, information, after the first section with the introduction. Please, structure this in this way or clearly state in “Organization” line 74 that these sections review needed theory.
  • Please, include a description of the algorithm, step by step.
  • In Fig. 7, and 6, I am not sure why in general your solutions have a “ripple effect”. In my view the simplest explanation (regularized) should be the option. You should explain why do you have this behavior.
  • It would be interesting to analyze the performanceas noise is reduced to cero (e.g. changing noise in line 289-291)
  • It is annoying that the truncated version be better than the original (standard) one. See Table 3. Why?
  • Notation is confusing: m as far as I understand is used for 1) mean in line 21, 2) number of terms in a series, e.g. line 121, 3) inducing points in line 201. In 2012 I am not sure of the meaning of N*. In line 231, M in M-dimensional is not defined, or should it be m?
  • There is no analysis of m, the number of needed basis functions. How does it behave as m increases? The number used, 25, is given at line 336, please, introduce it at the beginning of the experimental section.
  • A detailed analysis of the computational complexity compared to the sparse approaches should be provided. If you are using m=25 and 10 inducing points, please theoretically estimate the operations needed. Note that programming plays a central role and different software implementations could provide quite different results in time. Libraries used (lines 324-325) are different. Also, the sparse version is from 2005, and I am not sure if better approaches have been proposed that further reduce the computational complexity. We have also approaches based in the frequency domain.
  • I do not understand what you say about cache the result, in line 239.
  • Please, better explain, from a practical point of view, the impact of lines 225-226.

There is a typo at the end of line 304

Author Response

Comments 1: As far as I understand it is not until Section 5 that you introduce any new content. But you introduce even proofs. This is rare for a publication, where usually the second section includes all needed, not new, information, after the first section with the introduction. Please, structure this in this way or clearly state in “Organization” line 74 that these sections review needed theory. 

Response 1 : We thank the reviewer for this remark. In the “Organization” part of the revised version we state that the first three sections provide the necessary theoretical background.

Comments 2 : Please, include a description of the algorithm, step by step.

Response 2 : A full description of the algorithm has been added, please see Algorithm 1. 

Comments 3 : In Fig. 7, and 6, I am not sure why in general your solutions have a “ripple effect”. In my view the  simplest explanation (regularized) should be the option. You should explain why do you have this behavior.

Response 3 : Thank you for your valuable feedback. The "ripple effect" observed in Figures 6 and 7 is primarily due to two factors:

1.    Noisy dataset: The real datasets used in our experiments contains a significant level of noise. This naturally affects the fitting process, leading to fluctuations in the results.

2.    Polynomial basis: The oscillatory behavior is a consequence of using a polynomial basis for fitting. High-degree polynomials, especially in the presence of noise, tend to introduce oscillations, a well-known phenomenon often referred to as Runge’s phenomenon in interpolation.

Comments 4 : It would be interesting to analyze the performanceas noise is reduced to cero (e.g. changing noise in line 289-291)

Response 4 : We acknowledge that analyzing the performance as noise is reduced to zero is an interesting direction. In fact, we have conducted extensive experiments varying the noise level and observed that the Signal-to-Noise Ratio (SNR) decreases as the noise increases. These results confirm the expected trend and further validate our approach. However, due to space constraints and to maintain the paper's focus, we did not include these additional experiments in the manuscript.

Comments 5 : It is annoying that the truncated version be better than the original (standard) one. See Table 3. Why?

Response 5 : We understand your concern regarding the truncated version outperforming the original (standard) one in Table 3. However, this result is expected in our case. The truncation strategy is specifically designed to enhance efficiency in large-scale scenarios by reducing redundancy and mitigating overfitting, which can sometimes affect the standard approach. As a result, for big data settings, our method naturally performs better by focusing on the most relevant components while preserving accuracy. 

Comments 6 : Notation is confusing: m as far as I understand is used for 1) mean in line 21, 2) number of terms in a series, e.g. line 121, 3) inducing points in line 201. In 2012 I am not sure of the meaning of N*. In line 231, M in M-dimensional is not defined, or should it be m?

Response 6 : Thank you for pointing out the notation inconsistencies. We have now revised the notation as follows:

1.    The mean is now denoted as \mu.

2.    The number of terms in a series remains m.

3.    Inducing points are now represented by n.

Regarding your other concerns:

1.    In line 2012, N^* refers to N∖0.

2.    In line 231, we agree with you it should be "m-dimensional," and we have corrected this accordingly.

Comments 7 : There is no analysis of m, the number of needed basis functions. How does it behave as m increases? The number used, 25, is given at line 336, please, introduce it at the beginning of the experimental section.

Response 7 : The choice of the truncation order was determined based on a series of comprehensive empirical tests. We observed that increasing this number beyond a certain point negatively impacts prediction quality due to overfitting as for the canonical GP, while using a smaller number leads to underfitting. The chosen value represents a balance between these two effects and was selected based on experiments across different empirical datasets. We have now introduced this information at the beginning of the experimental results section for better clarity. 

Comments 8 : A detailed analysis of the computational complexity compared to the sparse approaches should be provided. If you are using m=25 and 10 inducing points, please theoretically estimate the operations needed. Note that programming plays a central role and different software implementations could provide quite different results in time. Libraries used (lines 324-325) are different. Also, the sparse version is from 2005, and I am not sure if better approaches have been proposed that further reduce the computational complexity. We have also approaches based in the frequency domain.

Response 8 : We appreciate the reviewer’s comments and would like to clarify the following points:

1.    Both the sparse Gaussian Process (GP) and our proposed Low-Complexity Gaussian Process (LCGP) have a computational complexity of O(Nn^2) for n  inducing points and O(Nm^2)  for m  basis functions, respectively. In our experiments, we set n = 10  and m = 25 , as these values were empirically found to be optimal, ensuring a fair comparison. A detailed analysis of the computational complexity of each method has now been provided in the paper. 

2.    The libraries used differ because the sparse GP implementation is not available in Scikit-learn. Instead, we relied on a separate implementation. Regarding the reference to the 2005 paper, we cited it as a pioneering work on sparse GPs, but our implementation is based on a more recent version developed in 2014.

3.    We have acknowledged approaches based on the frequency domain, such as the one proposed by Fritz et al. (2009). However, our comparison focuses on standard GP and sparse GP, as both methods aim to reduce complexity for both learning and inference. In contrast, frequency-domain approaches primarily improve efficiency during the learning phase, making them less directly comparable in our context.

Comments 9 : I do not understand what you say about cache the result, in line 239.

Response 9 : We mean that it is preferable to compute and store the result of \boldsymbol{\Phi}^\top \boldsymbol{\Phi} in memory once. We rectified it for the revised version. 

Comments 10 : Please, better explain, from a practical point of view, the impact of lines 225-226.

Response 10 : Thank you for your feedback. We have provided a more detailed explanation of the practical impact of lines 225-226 to clarify its implications. 

Comments 11 : There is a typo at the end of line 304

Response 11 : We have corrected the typo at the end of line 304. 

Reviewer 2 Report

Comments and Suggestions for Authors

please find my remarks in the attached pdf file

Author Response

Comments 1 : The paper is well written and the design is clear. The proposed method seems promising. The tests presented in Section 7 are convincing, but, in addition to testing the method with different operators, it would be useful to compare the result of the method with other standard methods that could serve as benchmark. Also, in the paper there is no definitive indication on how to select a priori the most effective orthogonal basis, even if the results for different basis are comparable in the presented tests.

Response 1 : We appreciate the reviewer’s comments. We agree that the proposed method could be compared with standard benchmark methods. However, our focus is on comparing it with approaches that reduce the complexity of both inference and learning. Additionally, the number of retained orthogonal basis functions, m, was empirically determined to be optimal based on our experiments.

Comments 2 : As a minor remark, it seems that the statement of Theorem 4 (lines 254 to 258) contains some errors.

- line 256: ”If ϕj is the eigenvector corresponding to the eigenvalue λj .” is an incomplete statement.

- line 258: ”eigenfunction still the same” should be ” eigenfunctions are still the same”.

Response 2 : Thank you for pointing this out. We have carefully reviewed and corrected the errors in the statement of Theorem 4 in the revised version.

Round 2

Reviewer 1 Report

Comments and Suggestions for Authors

Thank you for your responses to my comments and suggestions. 

My only main remaining concern is about complexity. As far as I understand from the reported results the proposal is by far more efficient (in time) than previous (Sparse) approaches. In the manuscript, you theoretically compare c2 x N x n^2 for n = 10  inducing points versus c1 x N x m^2 for m = 25 the number of terms. The computational time for c1 x N x m^2 is 1/100 the time for c2 x N x n^2 in the tables included. Is it c1 << c2 ? Otherwise, please explain.

Author Response

Comments 1 : My only main remaining concern is about complexity. As far as I understand from the reported results the proposal is by far more efficient (in time) than previous (Sparse) approaches. In the manuscript, you theoretically compare c2 x N x n^2 for n = 10 inducing points versus c1 x N x m^2 for m = 25 the number of terms. The computational time for c1 x N x m^2 is 1/100 the time for c2 x N x n^2 in the tables included. Is it c1 << c2 ? Otherwise, please explain.

Response 1 : We sincerely thank the reviewer for the insightful feedback and comments. To clarify this point, our proposed method is indeed more computationally efficient than Sparse GP, even though the number of inducing points (n = 10) is lower than the truncation order (m = 25). The theoretical complexities for inference are O(Nm^2) for our method and O(Nn^2) for Sparse GP, as they are both dominated by the number of operations required for each approach.

However, when considering the total computational time (measured in minutes) in Figure 8(c), the key difference lies in the complexity of the learning phase. While Sparse GP retains the O(Nn^2) complexity, our method benefits from a significantly lower learning complexity of O(m^3) (please see the detailed justification at the end of Section 5).

For instance, in our experimental setup, where N = k \times 1000 for k = 1,..., 12 , the complexity for Sparse GP scales as O(1000 \times 100), \dots, O(12000 \times 100), whereas for our method, it remains constant at O(25^3) for all values of k. This difference explains the exponential behavior observed for Sparse GP in Figure 8(c) and the quasi-constant computational time for our approach.